# Results of a Surface Roughness Comparison between Stylus Instruments and Confocal Microscopes

**DOI:** 10.3390/ma15165495

**Published:** 2022-08-10

**Authors:** Alberto Mínguez-Martínez, Piera Maresca, Jesús Caja, Jesús de Vicente y Oliva

**Affiliations:** 1Centro Láser, Universidad Politécnica de Madrid, Alan Turing 1, 28038 Madrid, Spain; 2Laboratorio de Metrología y Metrotecnia (LMM), Escuela Técnica Superior de Ingeniería Industrial, Universidad Politécnica de Madrid, José Gutiérrez Abascal, 2, 28006 Madrid, Spain; 3Escuela Técnica Superior de Ingeniería y Diseño Industrial, Universidad Politécnica de Madrid, Ronda de Valencia 3, 28012 Madrid, Spain

**Keywords:** spherical cap, confocal microscope, traceability, material standard, laser system, stylus instrument

## Abstract

This article presents the results of an LMM-R-2019 interlaboratory comparison. Such comparisons of different families of measuring instruments are one of the activities conducted among the calibration laboratories to maintain their ISO 17025 accreditation. Given that the study of surface roughness is becoming increasingly important in the field of dimensional metrology, the comparison focused on determining the Ra parameter on a pseudorandom metallic roughness standard using two types of measuring instruments: physical contact (stylus instruments) and optical (confocal microscopes). Among the aspects studied was whether the roughness measurements obtained using calibrated confocal microscopes could be compared with those using traditional methods since optical instruments obtain measurements more quickly and responsively than do stylus instruments. The results showed that roughness measurements using confocal microscopes are comparable with those from a traditional stylus instrument.

## 1. Introduction

### 1.1. Accreditation

All products are launched commercially on the market after exhaustive quality control testing, which includes dimensional control. Measurements in general, and dimensional measurements in particular, play a very important role in everyday life. Being able to carry out correct measurements is very important from a financial point of view since it provides interchangeability for products and services on a global level. For a company to have sufficient credibility and confidence in its products or services, it needs an independent evaluation system, also known as accreditation. This internationally established tool helps to instil confidence that a company has performed a test or calibration correctly, among other conformity assessment activities. Examples of advantages that accreditation provides include

Reliability and recognition for a company’s products and services,Lower possibility of providing a service or making a product with defects,Savings in the time and cost of re-evaluating products,Opening business opportunities in international markets by reducing costs for manufacturers and exporters.

At the national level, each country has an entity that carries out these conformity assessment activities independently and rigorously. This may be a National Metrology Institute in accordance with the document, *Quality Management Systems in the CIPM MRA-G-12* [1]. In Spain, the organization responsible for carrying out these tasks is the National Accreditation Body (ENAC) [2]. By having international agreements with different international organisations, such as European Accreditation (EA), International Laboratory Accreditation Cooperation (ILAC) and International Accreditation Forum (IAF), an ENAC certificate is recognised by the signatory countries of these agreements.

Among the different services it offers for calibration laboratories, ENAC assesses compliance with the requirements established in the ISO/IEC 17025 standard [3], which describes the general requirements for the competence, impartiality and operation of laboratories. Among the different activities for evaluating these requirements are interlaboratory comparisons (ILCs). According to Guthrie in [4], these are exercises conducted by a group of calibration laboratories to evaluate their performance and compare it with other laboratories. In accordance with ISO/IEC 17025 and ISO/IEC 17043 standards, each participating laboratory in an ILC carries out measurements on the same or similar objects to determine a physical property or feature according to previously agreed conditions [3,5]. Guthrie established that these activities could [4]

Evaluate random variation in the results of measurements conducted by different participating laboratories,Determine systematic differences in laboratory measurement results, andDetermine the value of one or more physical properties for one or more artefacts.

The ISO/IEC 17025 standard also establishes that non-standard methods, methods developed in a specific laboratory and standard methods used outside this laboratory that are not within the scope of accreditation can be validated [3].

### 1.2. Evaluation of Surface Roughness

As commented by Leach et al. in [6], the measurement of the geometric properties of parts is a crucial factor in determining the functional properties of each component, as well as controlling the manufacturing parameters. As Pawlus et al. established in [7,8], the surface topography is the result of manufacturing, functioning conditions, and wear. One of the properties that is becoming increasingly important in the manufacturing industry is the evaluation and analysis of surface texture according to the study carried out by Todhunter et al. in [9]. The surface of a part separates it from what is not the part whether it be the environment or another object, as established by Leach et al. [6,10]. As Nieslony et al. established in [11], it is crucial to distinguish between a mechanical and an electromagnetic surface as defined in the standard ISO 25178-2 [12]:Mechanical surface (Section 3.1.1.1 of [12]): “the boundary of the erosion, by spherical ball of radius r, of the locus of the centre of an ideal tactile sphere, also with radius r, rolled over the skin model of a workpiece”.Electromagnetic surface (Section 3.1.1.2 of [12]): “the surface obtained by the electromagnetic interaction with the skin model of a workpiece”.

According to the ISO 25178-6 standard [13] and as stated by Leach in [14], there are three types of methods to analyse surface texture:

Line-profiling: These provide a 2-dimensional plot or profile of the surface, expressed mathematically as z(x).Areal surface: These provide a 3-dimensional plot of the surface, expressed mathematically as z(x,y). Some measurement instruments that work with line profiling methods can work with areal surface methods, by extracting parallel profiles along the *y*-axis.Area-integrating: These methods provide a numerical result dependent on the area-integrating properties of the surface texture, by analysing the distribution of the scattered light.

Figure 1 of the ISO 25178-6 standard [13] specifies the different measuring instruments used to analyse a surface by each of the methods described above. Caja et al. in [15] and Townsend et al. in [16] established another classification according to the surface analysis method whether by mechanical contact or optical methods.

Contact stylus instruments: These mechanically contact the surface being scanned using a tip, the movement of which is converted into a signal as the function z(x). The z-coordinate of the contact point is determined by the transverse trajectory of the instrument axes, the shape of the tip, the deviation of the probe and the local inclination of the surface [6,13,14]. These instruments provide measurements of the mechanical surface of a sample.Optical instruments: These emit electromagnetic radiation reflected from the surface and detected by a sensor. The intensity of the detected radiation is used to calculate the spatial coordinates [6,14]. There are two types: those that measure the real surface of the study piece (using line-profiling or areal surface methods) and those that analyse scattered light (area-integrating methods). These instruments provide measurements of the electromagnetic surface of a sample. Among the different types of optical instruments are point autofocus, focus variation, interferometric (phase shifting and coherence), holographic digital microscopy, and confocal microscopes.

Mathia et al. also highlighted in [17] the importance of scanning tunnelling microscopes (STMs) and atomic force microscopes (AFMs) in the analysis of surface roughness. These instruments are usually used as scanning probe microscopes. There are other hybrid measuring instruments that allow the measurement of a surface with both optical and contact techniques. An example is the Infinite Focus Measurement (IFM) machine used by Nieslony et al. in [11].

Surface texture is quantified by a series of parameters established by mathematical analysis. As Królczyk et al. established in [18], the emerging aim in science is to determine functional surface parameters of the samples. Determination of the 2D roughness parameters is included in the ISO 4287 [19], ISO 12085 [20] and ISO 13565-2 [21] specification standards, while the 3D roughness parameters are included in the ISO 25178-2 standard [12]. Among the different standards, it is possible to find almost 300 parameters to characterize the surface, according to Mathia et al. in [17]. They could be classified into height, spatial, hybrid, functional, end and other, as Pawlus et al. commented in [8]. In this article, we focus on 2D roughness parameters.

According to the results of the survey conducted by Todhunter et al. in [9] and as Pawlus et al. introduced in [7], the most used 2D roughness parameter at the industrial level is the vertical parameter Ra (see Figure 1), which is calculated as
(1)$$Ra=\frac{1}{\ell_{p}}\int_{0}^{\ell_{p}}\left|z\left(x\right)\right|dx$$

As can be seen, for each evaluation length $l_{p}$ the part of the profile below the midline becomes positive and the arithmetic mean of the profile |z(x)| is calculated for that length. The advantage of this parameter is that it is affected less than other parameters (such as Rz, Rp, Rv, Rc and Rt) by scratches that the standard may have on its surface, foreign particles present in the groove, or by noise introduced during measurement.

Another widely used parameter in industrial environments is Rz (see Figure 2). It is calculated as the sum of height of the maximum profile peak height $Z_{p}$ and the maximum profile valley depth $\;Z_{v}$ [19], as detailed in Equation (2):(2)$$Rz=Z_{p}+Z_{v}$$

Note that this parameter is very sensitive to scratches, dirt, and other surface defects. As Caja et al. pointed out in [15], there are substantial differences between the analysis of a surface by physical contact with stylus instruments and by optical measuring instruments, due to the difference in the physical measurement principles of each. If the same surface is measured using different types of measuring instruments, the results are comparable, however different they may seem at first. According to Leach et al. in [22], optical measuring instruments present some potential advantages such as the absence of the risk of damage to the surface of the sample and a shorter time spent performing the measurement. However, optical measuring instruments historically have not always provided values equivalent to those obtained by contact stylus instruments. Leach et al. also point out in [22] that the problems presented by optical measuring instruments make necessary an a priori understanding of the nature of the surface. ILCs are very useful for these types of tasks. In this study, we show the ILC LMM-R-2019 surface roughness results in which the authors in 2019 participated with other nationally accredited laboratories. Among other things, the intention was to compare the measurements of the parameter Ra using two optical measuring instruments (confocal microscopes) with those obtained using stylus instruments from several ENAC accredited laboratories.

## 2. Interlaboratory Comparison LMM-R-2019

The following objectives were pursued in the ILC, LMM-R-2019:Study the performance of the participating laboratories (accredited if possible) by determining the Ra and Rz parameters for a roughness standard. These are the most used and studied roughness parameters according to the study of Todhunter et al. in [9].Determine the degree of compatibility among the different laboratory results. Section 2.4 details the procedure.Study the behaviour of optical measuring instruments with respect to contact methods. According to Leach et al. in [22], there are no widely accepted methods for calibrating optical measuring instrument. For this reason, this ILC could be used to validate the calibration procedures of the different confocal microscopes.

### 2.1. Participants

Two types of laboratories participated.

Accredited Calibration Laboratories (ACLs): A total of 7 ACLs participated from the public and industrial sectors and university centre laboratories. Only six laboratories in Spain are accredited by ENAC (the Spanish Accreditation Body) under ISO 17025 [3] to perform calibrations in roughness, and all of them participated in this ILC. All six are accredited to perform calibrations of roughness standards by determining Ra and Rz roughness parameters (amplitude parameters [19]). For this reason, the ILC was limited only to those two amplitude parameters. Only two of these six ACLs are accredited for the determination of Sm (spacing parameter [19]). The seventh ACL that participated in the ILC is accredited by ENAC in dimensional metrology but not roughness. Among them was the organiser of the ILC, the Metrology and Metrotechnics Laboratory (Laboratorio de Metrología y Metrotecnia, LMM) at Madrid Polytechnic University (Universidad Politécnica de Madrid, UPM).Research centres: The two participating research centres were the Dimensional Metrology Laboratory (LMD) and the Laser Centre (CL), both from UPM. It should be noted that the CL and LMM are inter-related, as they share staff. However, unlike the LMD, they do not share facilities.

All laboratories were traceable to the Spanish Metrology Centre (CEM). It was also remarkable that all of them performed roughness measurements in industrial environments. For this reason, the calibration and measuring procedures should be as simple and robust as possible.

### 2.2. Measuring Instruments

Two types of measuring instruments were used.

Stylus. All laboratories except CL used a stylus. These are classic measuring instruments on which all traceable and precision roughness measurements have been based so far. Their main advantage is that they are accepted worldwide for roughness measurement, and there is a lot of know-how based on its use. There are many well-established manufacturing processes that have quality assurance based on roughness measurements with these instruments. However, they have several limitations. As Leach et al. describe in [22], one of the most critical parameters is the shape of the stylus. Typically, it has a 2 µm radius tip, which may damage the surface of the samples and provide a too-coarse filter for certain applications.Optical: two confocal microscopes were used in the ILC LMM-R-2019. Confocal microscopy is an optical measurement method of surface topography in which the light reflected from the surface of the sample is filtered by a pinhole that discriminates what is in the focal plane from what is not according to ISO 25178-6 [13]. Mínguez-Martínez et al. have a schematic diagram of the filtering principle in Figure 2 of [23]. By taking images at different heights, a 3D reconstruction of the sample surface can be created using specialised software. These are optical measuring instruments the use of which is expanding as they make possible the carrying out of non-contact measurements of 3D roughness parameters. However, as stated by Leach et al. in [22], large differences have been observed in the results compared with those provided by traditional stylus instruments. In addition, all the acquired know-how and the procedures and techniques developed for stylus instruments may not work as well with these measuring instruments.

The instruments used in the two research centres were LEICA DCM3D confocal microscopes. The CL one was calibrated according to the procedure proposed by Mínguez-Martínez et al. in [23], while the procedure proposed by Wang et al. in [24] was used for the one in LMD. It is also necessary to say that the CL confocal microscope was not able to provide the Rz roughness parameter, so this measuring instrument only participated in the determination of Ra roughness parameter. Measurements were made of the object in Figure 3.

This is a metallic, pseudorandom roughness standard, HALLE brand, model A1, and the roughness parameter Ra was to be determined on its standard face. The nominal values of the standard are Ra=0.8 μm and Rz=4.5 μm. The measuring face is as indicated in the figure.

### 2.3. ILC Description

The organising laboratory, LMM, made an initial measurement and sent the standard to the second laboratory. From then on, each laboratory sent the standard to the next one, according to a previously established schedule. After receiving the standards, each participating laboratory had to send an email to the LMM with a report on the status of the standard, especially of its measurement surface. Finally, the standard reached the LMM which performed a second measurement.

Each measurement was made in the area indicated according to the following principles:The scan had to be located as close as possible to the measurement face axis (straight line collinear with the arrow engraved on the face).The evaluation length could not exceed the area enclosed by the dark marks engraved on the right and left of the measurement face.The evaluation length had to be centred with respect to the measurement face.

In accordance with the reference standard ISO 4288 [19], the following measurement conditions were established.

Evaluation length: $l_{p}=4\;\mathrm{mm}$.Basic sampling length: $l_{r}=0.8\;\mathrm{mm}$.Reference temperature: 20 °C.

As confocal microscopes cannot generally reach these $l_{p}$ and $l_{r}$ values, the lengths were required to be as close as possible. In addition, if the temperature could not be controlled, the LMM had to be informed to take this into account.

After the laboratories had performed the measurements, the results were recorded on a calibration certificate. This included the measurement result (the parameter Ra expressed in µm, to the appropriate number of significant figures), the expanded uncertainty (with coverage factor k=2, which corresponds to a coverage probability of ~95%), the values of $l_{p}$ and $l_{r}$ used and the measurement temperature. This was all done according to International Metrology Vocabulary (VIM) terminology [25]. All measurement uncertainties were calculated according to the ISO-GUM guide [26] or document EA-04/02 M:2013 [27], the standard procedures used by ACLs.

When all measurements were done, each participant emailed the others a protected zip file with the calibration certificate. When all participants confirmed receipt of the email to the MM, it sent another email to all the participants who sent the code to be able to decompress the zip file. This meant all participants could simultaneously access the calibration results.

The LMM prepared a final report for the ILC, LMM-R-2019, summarising the results from all the participants with different statistical parameters, which made it possible to compare the performance of each laboratory.

### 2.4. Estimation of the Standard Value and Statistical Parameters

This section establishes the procedure for estimating the measured standard value, which was assumed to be stable over time. The one used during the ILC can be described as a real number for which the estimate $x_{0}$ (with measurement uncertainty $u\left(x_{0}\right)$) had to be evaluated. In general, if the number of laboratories is n, the problem can be described by means of the vector equation described by Cox in [28]:(3)$$\vec{u}\cdot x_{0}=\left(\begin{array}{c} 1\\ \vdots \\ 1 \end{array}\right)\cdot x_{0}≅\left(\begin{array}{c} x_{1}\\ \vdots \\ x_{n} \end{array}\right)=\vec{x}$$

As all the laboratories were traceable to the CEM, the correlation among them was not zero, so the covariance matrix of the independent term $\vec{x}$ had to be calculated. Equation (4) describes how to calculate this matrix:(4)$$cov\left(x\right)=\overset{=}{C_{x}}=\left(\begin{array}{ccc} u^{2}\left(x_{1}\right) & \cdots & r_{1n}\cdot u\left(x_{1}\right)\cdot u\left(x_{n}\right)\\ \vdots & \ddots & \vdots \\ r_{n1}\cdot u\left(x_{n}\right)\cdot u\left(x_{1}\right) & \cdots & u^{2}\left(x_{n}\right) \end{array}\right)$$

It should be noted that

Each term $u\left(x_{i}\right)$ corresponds to the uncertainty obtained by each of the laboratories.Each term $r_{ij}$ corresponds to the correlation between laboratories i and j.$0\leq r_{ij}\leq 1$.$r_{ij}=r_{ji}$.As all ACLs are traceable to the CEM, the correlation among independent laboratories was considered to be $r_{ij}=0.1$.As two sets of measurements were performed in the LMM (at the beginning and end of the ILC) using the same equipment under similar conditions and with the same technician, the correlation between them must be the highest of all. Therefore, the value r=0.6 was assigned.As two measurements were made in the LMD (one with a stylus instrument and the other with a confocal microscope), the correlation between these two measurements must be slightly higher than those with other laboratories but not as high as the two measurements made by the LMM. Therefore, the value r=0.5 was assigned.For measurements made in the CL, as it was related to the LMM via its technicians, the correlation among them and those made in the LMM must be higher than between the CL and other laboratories but less than those of the LMD. Therefore, the value r=0.4 was assigned.

It can be shown that the solution to the matrix problem can be reduced to the following expressions:(5)$$x_{0}=\frac{\vec{u^{T}}\cdot {\overset{=}{C_{x}}}^{-1}\cdot \vec{x}}{\vec{u^{T}}\cdot {\overset{=}{C_{x}}}^{-1}\cdot \vec{u}}$$
(6)$$U\left(x_{0}\right)=\frac{1}{\vec{u^{T}}\cdot {\overset{=}{C_{x}}}^{-1}\cdot \vec{u}}$$

Once the measured value of the standard was estimated, the statistical parameters were calculated to evaluate the performance of each laboratory, which is included in the reference standard ISO 17043 [5]. Please note that due to the correlation between $x_{i}$ and reference value $x_{0}$ in Equation (9) there is a negative sign in the denominator instead of the usual positive sign when $x_{i}$ and $x_{0}$ are statistically independent.
(7)$$\mathrm{Differences}:\;∆x_{i}=x_{i}-x_{0}$$
(8)$$\mathrm{Standard}\;\mathrm{uncertainty}\;\mathrm{of}\;\mathrm{differences}:\;u\left(∆x_{i}\right)=\sqrt{u^{2}\left(x_{i}\right)-u^{2}\left(x_{0}\right)}$$
(9)$$\mathrm{Normalized}\;\mathrm{error}\;\mathrm{or}\;\mathrm{Compatibility}\;\mathrm{Index}:\;E_{n}=\frac{x_{i}-x_{0}}{\sqrt{u^{2}\left(x_{i}\right)-u^{2}\left(x_{0}\right)}}$$

## 3. Results

Table 1 shows the Ra values (expressed as $x_{i}$) and the associated expanded uncertainty (with coverage factor k=2, that provides a coverage probability of 95%) obtained by each of the participating laboratories. All units are in µm.

As can be seen, the results are very close to the nominal values of the material standard (Ra=0.8 μm and Rz=4.5 μm). Of all the results, we considered that the measurements made with the confocal microscopes and the estimation of the uncertainty associated with these measurements should be highlighted.

Attending to the Ra roughness parameter results, confocal microscopes did not seem to present significant deviations from the laboratories that used stylus instruments, considering the uncertainty. This meant that the deviations were lower than the uncertainty. For example, the maximum deviation was between the confocal microscope of CL and Lab 2, Lab 4 and Lab 5 (0.09 µm), which was lower than the uncertainty of the CL and very similar to that of Lab2, Lab 4 and Lab 5. It is also important to highlight that the confocal microscope’s uncertainties (0.12 and 0.16 µm) are clearly higher than the uncertainties of the stylus instruments (by approximately 0.08 μm on average), that is, between 50 and 100% higher.

Attending to Rz roughness parameter results, the maximum variability observed was 0.44 μm. It was important to consider that this parameter was more statistically unstable, which is the reason that greater variability is observed in relative terms. The maximum difference between stylus instrument and confocal microscope measurements is 0.34 µm, which is clearly lower than the uncertainty of the confocal microscope (0.48 µm) and is on the order of most uncertainties obtained with the stylus instruments. For this reason, it could be said that the uncertainty for confocal microscopes is similar to those for the stylus instruments.

The measurements for the CL confocal microscope were made with the 10× objective, the same with which the confocal microscope was calibrated. This objective provided a distance between planes (voxel height) of 2 µm. Measurements were taken so that the roughness parameter Ra could be measured both along the instrument *x*- and *y*-axes (see Figure 4).

The confocal microscope provides 3D images like the one in Figure 5.

## 4. Discussion

### 4.1. Ra Roughness Parameter

The measurement uncertainty of the equipment used by the CL was estimated from the formula described by Mínguez-Martínez et al. in [23]:(10)$$U\left(Ra\right)=k\cdot \sqrt{{\bar{R}}^{2}\cdot u^{2}\left(c_{z}\right)+\frac{s^{2}\left(\bar{R}\right)}{m}+u^{2}\left(\bar{b}\right)+u^{2}\left(\delta b\right)+u_{noise}^{2}}$$
where $\bar{R}$ is the mean Ra measurement; $u\left(c_{z}\right)$ is the uncertainty associated with the z-axis scale correction factor ($c_{z}$); $s\left(\bar{R}\right)$ is the repeatability; m is the number of measurements made; $u\left(\bar{b}\right)$ is the uncertainty associated with the positive bias of the measurement instrument; u(δb) is the uncertainty associated with the variation in the positive bias of the measurement instrument; and $u_{noise}$ is the uncertainty component due to instrument noise. In this case, the value $u_{noise}=0$ was assigned as it is small enough to be considered negligible compared to the height of the voxel.

Similarly, the estimation of uncertainties of the equipment used by the LMD—Conf considers the following sources of uncertainty [24]:
Measurement repeatability,Z-axis calibration coefficient,Measurement noise (evaluated when measuring at standard flatness),Variation in light intensity used by the equipment, andVariation in the sample orientation (lack of perpendicularity of the sample with respect to the equipment optical axis).

Considering the results in Table 1, the correlation coefficients $r_{ij}$ and Equation (4), the covariance matrix is calculated as:
(11)$$\overset{=}{C_{x}}=(\begin{array}{ccccccccccc} 2.0250 & 0.2025 & 0.1800 & 0.1575 & 0.1800 & 0.1800 & 0.1800 & 0.2025 & 0.2700 & 1.4183 & 1.2150\\ 0.2025 & 2.0250 & 0.1800 & 0.1575 & 0.1800 & 0.1800 & 0.1800 & 0.2025 & 0.2700 & 0.3546 & 0.2025\\ 0.1800 & 0.1800 & 0.1600 & 0.1400 & 0.1600 & 0.1600 & 0.1600 & 0.1800 & 0.2400 & 0.3152 & 0.1800\\ 0.1575 & 0.1575 & 0.1400 & 1.2250 & 0.1400 & 0.1400 & 0.1400 & 0.1575 & 0.2100 & 0.2758 & 0.1575\\ 0.1800 & 0.1800 & 0.1600 & 0.1400 & 0.1600 & 0.1600 & 0.1600 & 0.1800 & 0.2400 & 0.3152 & 0.1800\\ 0.1800 & 0.1800 & 0.1600 & 0.1400 & 0.1600 & 0.1600 & 0.1600 & 0.1800 & 0.2400 & 0.3152 & 0.1600\\ 0.1800 & 0.1800 & 0.1600 & 0.1400 & 0.1600 & 0.1600 & 0.1600 & 0.1800 & 0.2400 & 0.3152 & 0.1600\\ 0.2025 & 0.2025 & 0.1800 & 0.1575 & 0.1800 & 0.1800 & 0.1800 & 2.0250 & 1.3500 & 0.3546 & 0.2025\\ 0.2700 & 0.2700 & 0.2400 & 0.2100 & 0.2400 & 0.2400 & 0.2400 & 1.3500 & 3.6000 & 0.4728 & 0.2700\\ 1.4183 & 0.3546 & 0.3152 & 0.2758 & 0.3152 & 0.3152 & 0.3152 & 0.3546 & 0.4728 & 6.2087 & 1.4183\\ 1.2150 & 0.2025 & 0.1800 & 0.1575 & 0.1800 & 0.1800 & 0.1800 & 0.2025 & 0.2700 & 1.4183 & 2.0250 \end{array})\cdot 10^{-3}$$

In this case, since all $r_{ij}\neq 0$, the matrix has all terms. The values are generally less than 1. Exceptions are for those laboratories where the correlation $r_{ij}>0.1$.

Using this matrix and Equations (5) and (6), $x_{0}$ and $U\left(x_{0}\right)$, respectively, can be calculated:(12)$$x_{0}=\frac{\vec{u^{T}}\cdot {\overset{=}{C_{x}}}^{-1}\cdot \vec{x}}{\vec{u^{T}}\cdot {\overset{=}{C_{x}}}^{-1}\cdot \vec{u}}=0.8264\;\mu m$$
(13)$$U\left(x_{0}\right)=\frac{1}{\vec{u^{T}}\cdot {\overset{=}{C_{x}}}^{-1}\cdot \vec{u}}=0.0372\;\mu m$$

The value of $x_{0}$ is slightly higher than the certified value; however, it is contained in the interval $\left(x_{0}-U\left(x_{0}\right),x_{0}+U\left(x_{0}\right)\right)$. Based on the values of Equations (12) and (13) and those of Table 1, the graph in Figure 6 was obtained. The blue point represents the value of each $x_{i}$ from Table 1; the vertical bars represent the uncertainty $U\left(x_{i}\right)$ from Table 1; the continuous orange line represents the $x_{0}$ value calculated with Equation (12); and the discontinuous orange lines represent the maximum and minimum values of $U\left(x_{0}\right)$ calculated with Equation (13).

### 4.2. Rz Roughness Parameter

Analogous to what has been done in Section 4.1, the covariance matrix was calculated:
(14)$$\overset{=}{C_{x}}=(\begin{array}{cccccccccc} 13.2250 & 2.9325 & 3.9100 & 2.0700 & 1.9550 & 2.2425 & 2.6450 & 2.2425 & 2.7600 & 7.9350\\ 2.9325 & 65.0250 & 8.6700 & 4.5900 & 4.3350 & 4.9725 & 5.8650 & 4.9725 & 6.1200 & 2.9325\\ 3.9100 & 8.6700 & 115.6000 & 6.1200 & 5.7800 & 6.6300 & 7.8200 & 6.6300 & 8.1600 & 3.9100\\ 2.0700 & 4.5900 & 6.1200 & 32.400 & 3.0600 & 3.5100 & 4.1400 & 3.5100 & 4.3200 & 2.0700\\ 1.9550 & 4.3350 & 5.7800 & 3.0600 & 28.9000 & 3.3150 & 3.9100 & 3.3150 & 4.0800 & 1.9550\\ 2.2425 & 4.9725 & 6.6300 & 3.5100 & 3.3150 & 38.0250 & 4.4850 & 3.8025 & 4.6800 & 2.2425\\ 2.6450 & 5.8650 & 7.8200 & 4.1400 & 3.9100 & 4.4850 & 52.9000 & 4.4850 & 5.5200 & 2.6450\\ 2.2425 & 4.9725 & 6.6300 & 3.5100 & 3.3150 & 3.8025 & 4.4850 & 38.0250 & 23.4000 & 2.2425\\ 2.7600 & 6.1200 & 8.1600 & 4.3200 & 4.0800 & 4.6800 & 5.5200 & 23.4000 & 57.6000 & 2.7600\\ 7.9350 & 2.9325 & 3.9100 & 2.0700 & 1.9550 & 2.2425 & 2.6450 & 2.2425 & 2.7600 & 13.2250 \end{array})\cdot 10^{-3}$$

In addition, since all $r_{ij}\neq 0$, the matrix has all terms. The values are generally less than 1 except for those laboratories where the correlation $r_{ij}>0.1$. Using this matrix and Equations (5) and (6), $x_{0}$ and $U\left(x_{0}\right)$, respectively, can be calculated:(15)$$x_{0}=\frac{\vec{u^{T}}\cdot {\overset{=}{C_{x}}}^{-1}\cdot \vec{x}}{\vec{u^{T}}\cdot {\overset{=}{C_{x}}}^{-1}\cdot \vec{u}}=4.3985\;\mu m$$
(16)$$U\left(x_{0}\right)=\frac{1}{\vec{u^{T}}\cdot {\overset{=}{C_{x}}}^{-1}\cdot \vec{u}}=0.1570\;\mu m$$

The value of $x_{0}$ is slightly higher than the certified value; however, it is contained in the interval $\left(x_{0}-U\left(x_{0}\right),x_{0}+U\left(x_{0}\right)\right)$. Taking into account the values of Equations (12) and (13) and those of Table 1, the graph in Figure 7 is obtained. The blue point represents the value of each $x_{i}$, from Table 1, the vertical bars represent the uncertainty $U\left(x_{i}\right)\;$ from Table 1; the continuous orange line represents the $x_{0}$ value, calculated with Equation (15); and the discontinuous orange lines represent the maximum and minimum values of $U\left(x_{0}\right)$ calculated with Equation (16).

As can be seen, in all cases, the uncertainty bars cover the interval of the average values of the measurand $\left(x_{0}-U\left(x_{0}\right),x_{0}+U\left(x_{0}\right)\right)$. Therefore, the measurement results from the different participating laboratories can be said to be compatible with each other. The result that deviates the most is the one obtained using the CL confocal microscope although this is still compatible with the other laboratories.

The calculation results for the statistical parameters from Equations (7)–(9) are shown in Table 2.

According to the results in Table 2, it is necessary to highlight that the maximum difference between $x_{0}$ and the different $x_{i}$ is 9.68% for Ra and 6.37% for Rz. If we consider the averages of the different $x_{i}$, the difference with $x_{0}$ is 2.4% for Ra and 3.1% for Rz. As can be seen, the maximum deviation between the average value and that obtained by each of the participating laboratories was 0.08 µm, which is that of the CL confocal microscope. However, the average value $x_{0}$ is in the range $\left(x_{i}-U\left(x_{i}\right),x_{i}+U\left(x_{i}\right)\right)$ for this laboratory. It also can be seen that the measurement uncertainties obtained with confocal microscopes are higher than those for stylus instruments. Theoretically, this result seems to be negative. However, the most important reading is that comparable results were found when using two measurement instruments having completely different measurement principles. This result is important since the use of optical measurement instruments is simpler than that of stylus instruments and can lead to savings for ACLs in money, time and effort.

In the end, the most important analysis corresponded to the $E_{n}$ number (sometimes known as “normalized error” or “compatibility index”). According to ISO 17043 [5], when $\left|E_{n}\right|\leq 1$, the result can be considered satisfactory. Therefore, all results from Table 1 for Ra or for Rz are satisfactory, especially those obtained with confocal microscopes.

## 5. Conclusions

The procedure followed for this ILC roughness study, LMM-R-2019, is submitted in this report. It asked participating accredited laboratories research centres to determine the surface roughness of a measurement standard from the vertical parameters Ra and Rz. In this ILC, the participating laboratories used both mechanical contact (stylus) and optical (confocal microscopes) measurement instruments.

The results were very good, and showed that the results from the different laboratories were compatible with each other. The largest deviation was between a confocal microscope and one of the laboratories that used a stylus instrument. The mean square of the deviation $E_{n}$ turned out to be 0.33 in the case of the Ra roughness parameter and 0.32 in the case of the Rz roughness parameter. If the laboratories had accurately estimated their uncertainties, the root mean square value would have been close to 0.5.

This allowed us to say that they seemed to be working correctly within a wide safety margin. In other words, the uncertainties of these laboratories were probably on the order of 0.06 µm while they were providing clearly higher uncertainties.

Finally, special mention should be made of the results from a confocal microscope. It is known that the measurement of roughness with optical instruments have significant biases compared to measurements made with traditional contact stylus instruments. However, in this comparison, the measurements made with confocal microscopes were fully compatible with those made with contact stylus instruments. The problem with the microscopes, may have been due in many cases to an incorrect calibration or one that was not focused on carrying out roughness measurements. In this case, it seems that the confocal instruments used were correctly calibrated and their uncertainties reasonably well estimated, resulting in compatibility with the results from contact stylus instruments.

This ILC was designed to measure one kind of material standard. In the future, we will repeat the measurements using another material standard (e.g., isotropic and anisotropic/periodic and random) with different roughness values.

## Figures and Tables

**Figure 1 materials-15-05495-f001:**
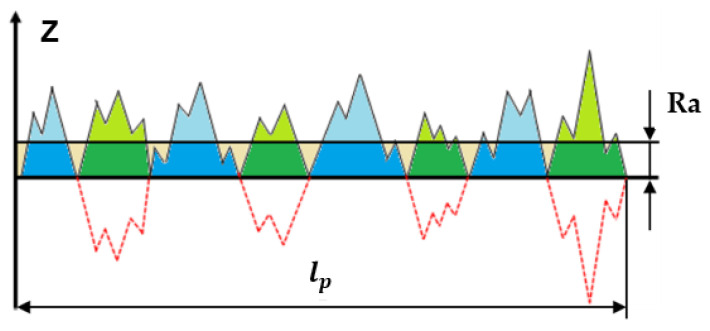
Geometric description of the roughness parameter Ra.

**Figure 2 materials-15-05495-f002:**
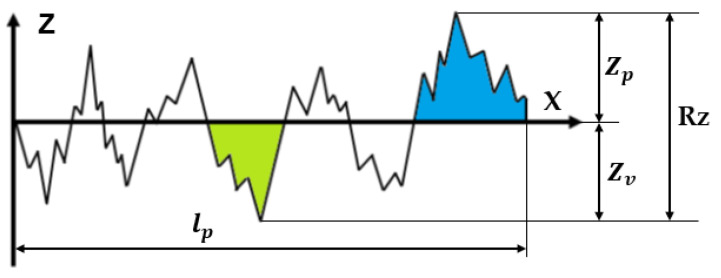
Geometric description of the roughness parameter Rz.

**Figure 3 materials-15-05495-f003:**
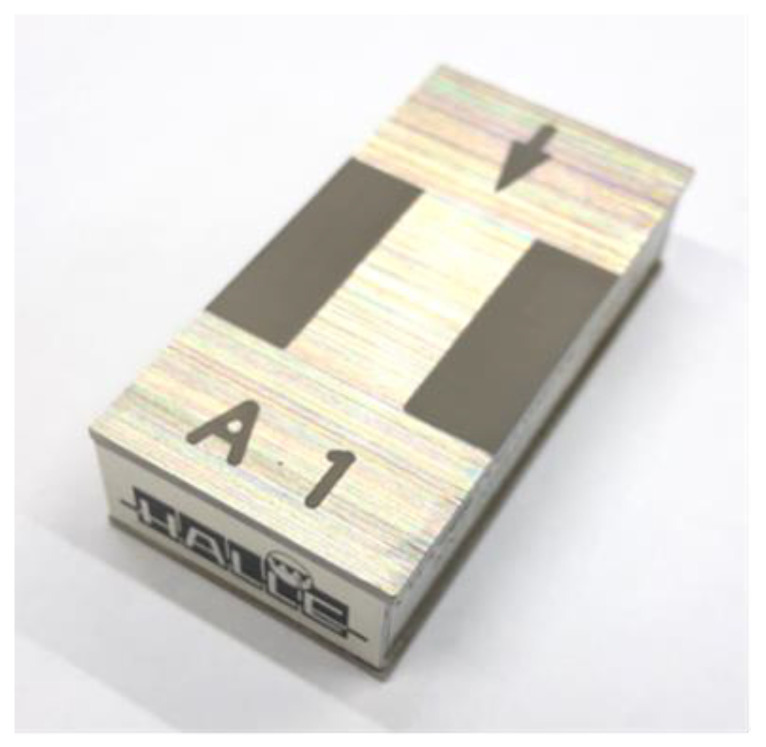
Roughness standard used in the ILC, LMM-R-2019.

**Figure 4 materials-15-05495-f004:**

Measurements with the CL confocal microscope. The red line corresponds to the extracted profile.

**Figure 5 materials-15-05495-f005:**
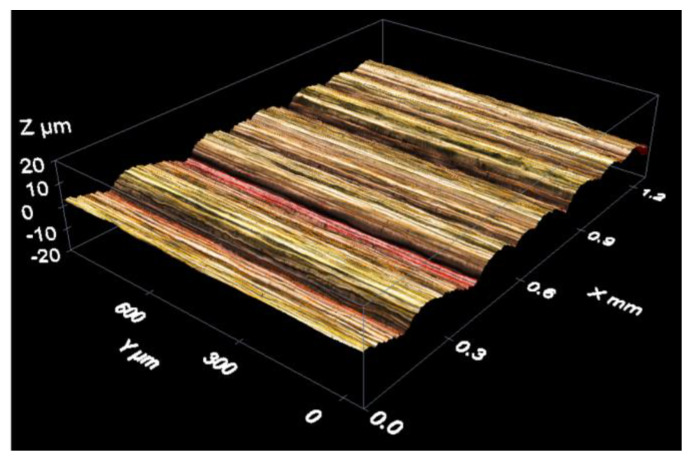
3D reconstruction of the standard surface measured with a confocal microscope.

**Figure 6 materials-15-05495-f006:**
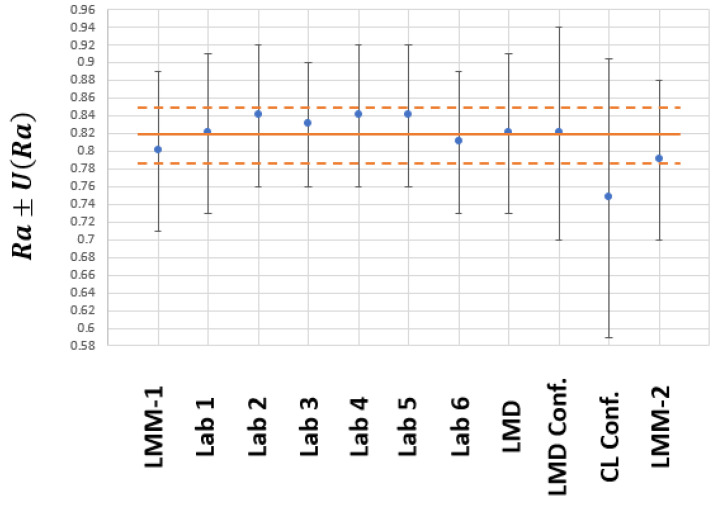
ILC results for Ra roughness parameter.

**Figure 7 materials-15-05495-f007:**
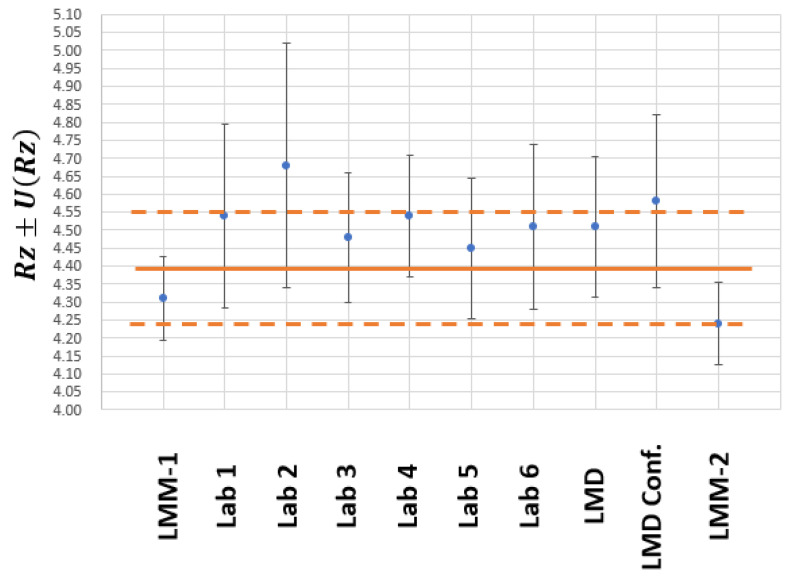
ILC results for Rz roughness parameter.

**Table 1 materials-15-05495-t001:** Measuring results from each participant laboratory for Ra and Rz roughness parameters.

		LMM-1	Lab1	Lab2	Lab3	Lab4	Lab5	Lab6	LMD	LMD-Conf.	CL-Conf.	LMM-2
#		1	2	3	4	5	6	7	8	9	10	11
**Ra**	$x_{i}$	0.80	0.82	0.84	0.83	0.84	0.84	0.81	0.82	0.82	0.75	0.79
	$U\left(x_{i}\right)$	0.09	0.09	0.08	0.07	0.08	0.08	0.08	0.09	0.12	0.16	0.09
**Rz**	$x_{i}$	4.31	4.54	4.68	4.48	4.54	4.45	4.51	4.51	4.58	-	4.24
	$U\left(x_{i}\right)$	0.23	0.51	0.68	0.36	0.34	0.39	0.46	0.39	0.48	-	0.23

**Table 2 materials-15-05495-t002:** Statistical parameters.

		LMM−1	Lab1	Lab2	Lab3	Lab4	Lab5	Lab6	LMD	LMD-Conf.	CL-Conf.	LMM-2
Ra	$∆x_{i}$ (µm)	−0.03	−0.01	0.01	0.00	0.01	0.01	−0.02	−0.01	−0.01	−0.08	−0.04
	$U\left(∆x_{i}\right)$ (µm)	0.08	0.08	0.07	0.06	0.07	0.07	0.07	0.08	0.11	0.16	0.08
	$E_{n}$	−0.32	−0.08	0.19	0.06	0.19	0.19	−0.23	−0.08	−0.06	−0.49	−0.44
Rz	$∆x_{i}$ (µm)	−0.09	0.14	0.28	0.08	0.14	0.05	0.11	0.11	0.18	−	−0.16
	$U\left(∆x_{i}\right)$ (µm)	0.17	0.49	0.66	0.32	0.30	0.36	0.43	0.36	0.45	−	0.17
	$E_{n}$	−0.53	0.29	0.43	0.25	0.47	0.14	0.26	0.31	0.4	−	−0.94

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
