# Peer review of "Results of a Surface Roughness Comparison between Stylus Instruments and Confocal Microscopes"

_materials, 2022, doi:10.3390/ma15165495_

Round 1

Reviewer 1 Report

Dear Authors,

please improve the paper according to the comments

1. the labels, symbols in figure 1 are not compatible with formula 1. fix it

2. comparative research of different types of measuring instruments is a well-known topic. the paper should emphasize the research gap and scientific significance of the presented research

3, doing research on a one sample is unreliable. I suggest extending the research to include surfaces after different processing with different functional properties characterized by different values of roughness parameters. Surfaces should be isotropic and anisotropic, periodic and random, one-process and two-process with different roughness heights. 

4. evaluation of only one parameter seems to be insufficient. please consider the evaluation of surface topography parameters. The evaluation of the Ra parameter often does not reflect the nature of the distribution of irregularities

5. why such instruments were used - please explain the advantages and disadvantages in comparison with each other, the benefits of using particular devices

6. explain in more detail the differences of the values shown in Table 1 in relation to nominal values

7. add a list of abbreviations 

8. improve the neatness of the work - the description of Table 1 and many typos in the text

Best regards 

Reviewer

Reviewer 2 Report

The paper has a good SoA and the study of surface roughness is becoming increasingly important in the the field of dimensional metrology, so is it is interesting and a good contribution in the field.

The paper is clear and well writen. The objectives of the study are explicit from the beginning and teh SoA is well done and acceptable.

I would suggest to include reference [4] right after "According to Guthrie" line 55, but it is acceptable in the end of the sentence. It get "more" readable such as in the sentence "As commented by Leach et al. in [6], the measurement..." line 70.

The methodology presented is claimed by authors as based in a good expertise in the area, but they demonstrate quite well through the paper the qualities of the approach. They referred VIM, ISO-GUM guide and EA-04/02 M:2013 for example, which are adequate.

The conclusions are supported by the results, despite legend of "Figure 5. ILC results." would greatly profit of a more extended description.Please carefully review the others. The tables are readable and authors presented the fundaments of the results and their focus.

After those minnor revisions, in my opinion the paper is acceptable to publication.

Round 2

Reviewer 1 Report

Dear Authors,

Thank you for improving your paper about my comments.

Best regards

Reviewer

Author Response

Dear reviwer,

Please find the new version of the article after minor revision.

Thank you again for your suggestions. Best regards,

The authors